METHODS

# `rtestim`: Time-varying reproduction number estimation with trend filtering

**Jiaping Liu**[1]*, **Zhenglun Cai**[2], **Paul Gustafson**[1], **Daniel J. McDonald**[1]

**1** Department of Statistics, The University of British Columbia, Vancouver, British Columbia, Canada,
**2** Centre for Health Evaluation and Outcome Sciences, The University of British Columbia, Vancouver, British Columbia, Canada

* jiaping.liu@stat.ubc.ca

**Data Availability Statement:** Data and computational code are available through the R package "rtestim" (https://dajmcdon.github.io/rtestim/) and the Github repository "rt-est-

## Abstract

To understand the transmissibility and spread of infectious diseases, epidemiologists turn to estimates of the instantaneous reproduction number. While many estimation approaches exist, their utility may be limited. Challenges of surveillance data collection, model assumptions that are unverifiable with data alone, and computationally inefficient frameworks are critical limitations for many existing approaches. We propose a discrete spline-based approach that solves a convex optimization problem—Poisson trend filtering—using the proximal Newton method. It produces a locally adaptive estimator for instantaneous reproduction number estimation with heterogeneous smoothness. Our methodology remains accurate even under some process misspecifications and is computationally efficient, even for large-scale data. The implementation is easily accessible in a lightweight R package `rtestim`.

## Author summary

Instantaneous reproduction number estimation presents many challenges due to data collection, modelling assumptions, and computational burden. Our motivation is to develop a model that produces accurate estimates, is robust to model misspecification, is straightforward to use, and is computationally efficient, even for large counts and long time periods. We propose a convex optimization model with an $\ell_1$ trend filtering penalty. It couples accurate estimation of the instantaneous reproduction number with desired smoothness. We solve the optimization using the proximal Newton method, which converges rapidly and is numerically stable. Our software, conveniently available in the R package `rtestim`, can produce estimates in seconds for incidence sequences with hundreds of observations. These estimates are produced for a sequence of tuning parameters and can be selected using a built-in cross validation procedure.

manuscript" (https://github.com/jiapivialiu/rt-est-manuscript).

**Funding:** DJM was partially supported by Centers for Disease Control and Prevention (under U011P001121 and 75D30123C15907) with url (https://www.cdc.gov) and National Sciences and Engineering Research Council (under ALLRP 581756-23 and RGPIN 2021-02618) with url (https://www.nserc-crsng.gc.ca). PG was partially supported by National Sciences and Engineering Research Council (under RGPIN 2019-03957) with url (https://www.nserc-crsng.gc.ca). The funders had no role in study design, data collection and analysis, decision to publish, or preparation of the manuscript.

# 1 Introduction

The effective reproduction number is defined to be the expected number of secondary infections produced by a primary infection where some part of the population is no longer susceptible. The effective reproduction number is a key quantity for understanding infectious disease dynamics including the potential size of an outbreak and the required stringency of control measures [1, 2]. The instantaneous reproduction number is a type of effective reproduction number that tracks the number of secondary infections at time $t$ relative to all preceding primary infections. This contrasts with the case reproduction number at $t$ which indexes a primary infection at time $t$, tracking the infectiousness of the cohort [3]. Tracking the time series of the effective reproduction number quantity is useful for understanding whether or not future infections are likely to increase or decrease from the current state [4]. Our focus is on the instantaneous reproduction number at time $t$, which we will denote $\mathcal{R}(t)$. Practically, as long as $\mathcal{R}(t) < 1$, infections will decline gradually, eventually resulting in a disease-free equilibrium, whereas when $\mathcal{R}(t) > 1$, infections will continue to increase, resulting in endemic equilibrium. While $\mathcal{R}(t)$ is fundamentally a continuous time quantity, it can be related to data only at discrete points in time $t = 1, \ldots, n$. This sequence of instantaneous reproduction numbers over time is not observable, but, nonetheless, is easily interpretable and describes the course of an epidemic. Therefore, a number of procedures exist to estimate $\mathcal{R}_t$ from different types of observed incidence data such as cases, deaths, or hospitalizations, while relying on various domain-specific assumptions, e.g., [5–8]. Importantly, accurate estimation of instantaneous reproduction numbers relies heavily on the quality of the available data, and, due to the limitations of data collection, such as underreporting and lack of standardization, estimation methodologies rely on various assumptions to compensate. Because model assumptions may not be easily verifiable from data alone, it is also critical for any estimation procedure to be robust to model misspecification.

Many existing approaches for instantaneous reproduction number estimation are Bayesian: they estimate the posterior distribution of $\mathcal{R}_t$ conditional on the observations. One of the first such approaches is the software `EpiEstim` [9], described by Cori et al. [10]. This method is prospective, focusing on the instantaneous reproduction number, and using only observations available up to time $t$ in order to estimate $\mathcal{R}_t$ for each $i = 1, \ldots, t$. An advantage of `EpiEstim` is its straightforward statistical model: new incidence data follows the Poisson distribution conditional on past incidence combined with the conjugate gamma prior distribution for $\mathcal{R}_t$ with fixed hyperparameters. Additionally, the serial interval distribution, the distribution of the period between onsets of primary and secondary infections in a population, is fixed and known. For this reason, `EpiEstim` requires little domain expertise for use, and it is computationally fast. Thompson et al. [11] modified this method to distinguish imported cases from local transmission and simultaneously estimate the serial interval distribution. Nash et al. [12] further extended `EpiEstim` by using "reconstructed" daily incidence data to handle irregularly spaced observations.

Recently, Abbott et al. [13] proposed a Bayesian latent variable framework, `EpiNow2` [14], which leverages incident cases, deaths or other available streams simultaneously along with allowing additional delay distributions (incubation period and onset to reporting delays) in modelling. Lison et al. [15] proposed an extension that handles missing data by imputation followed by a truncation adjustment. These modifications are intended to increase accuracy at the most recent (but most uncertain) timepoints, to aid policymakers. Parag et al. [16] also proposed a Bayesian approach, `EpiFilter`, based on the (discretized) Kalman filter and smoother. `EpiFilter` also estimates the posterior of $\mathcal{R}_t$ given using a Markov model for $\mathcal{R}_t$ and Poisson distributed incident cases. Compared to `EpiEstim`, however, `EpiFilter`

estimates $\mathcal{R}_t$ retrospectively using all available incidence data both before and after time $t$, with the goal of being more robust in low-incidence periods. Gressani et al. [17] proposed a Bayesian P-splines approach, EpiLPS, that assumes negative binomial distributed observations, allowing for overdispersion in the observed incidence. Trevisin et al. [18] also proposed a Bayesian model estimated with particle filtering to incorporate spatial structures. Bayesian approaches estimate the posterior distribution of the instantaneous reproduction numbers and possess the advantage that credible intervals may be easily computed. They also can incorporate prior knowledge on parameters. Another potential advantage is that a relatively large prior on the mean of $\mathcal{R}_t$ can be used to guard against erroneously concluding that an epidemic is shrinking [11]. However, a downside is that the induced bias can persist for long periods of time. Bayesian approaches that do not use conjugate priors, or that incorporate multilevel modelling, can be computationally expensive, especially when observed data sequences are long or hierarchical structures are complex, e.g., [13].

There are also frequentist approaches for $\mathcal{R}_t$ estimation. Abry et al. [19] proposed regularizing the smoothness of $\mathcal{R}_t$ through penalized regression with second-order temporal regularization, additional spatial penalties, and with Poisson loss. Pascal et al. [20] extended this procedure by adding a penalty on outliers. Pircalabelu et al. [21] proposed a spline-based model relying on the assumption of exponential-family distributed incidence. Ho et al. [22] estimated $\mathcal{R}_t$ while monitoring the time-varying level of overdispersion. There are other spline-based approaches such as [23, 24], autoregressive models with random effects [25] that are robust to low incidence, and generalized autoregressive moving average models [26] that are robust to measurement errors in incidence data.

We propose an instantaneous reproduction number estimator, that requires only incidence data. Our model makes the conditional Poisson assumption, similar to much of the prior work described above, but is empirically more robust to misspecification. This estimator is defined by a convex optimization problem with Poisson loss and $\ell_1$ penalty on the temporal evolution of $\log(\mathcal{R}_t)$ to impose smoothness over time. As a result, it generates discrete splines, and the estimated curves (on the logarithmic scale) appear to be piecewise polynomials of an order selected by the user. Importantly, the estimates are locally adaptive, meaning that different time ranges may possess heterogeneous smoothness. Because we penalize the logarithm of $\mathcal{R}_t$, we naturally accommodate the positivity requirement, in contrast to related methods [19, 20], can handle large or small incidence measurements, and are automatically (reasonably) robust to outliers without additional constraints (a feature of the $\ell_1$ penalty). A small illustration using three years of Covid-19 case data in Canada [27] is shown in Fig 1, where we use a time-varying serial interval distribution. The implementation is easily accessible in a lightweight R package called rtestim.

While our approach is straightforward and requires little domain knowledge for implementation, we also implement a number of refinements:

- the algorithm solves over a range of tuning parameters simultaneously, using warm starts to speed up subsequent solutions;

- cross-validation is built in (and used in all analyses below) to automatically select tuning parameters;

- parametric (gamma), non-parametric (any discretized delay), and time-varying delay distributions are allowed;

- irregularly spaced incidence data are easily accommodated;

- approximate confidence intervals for $\mathcal{R}_t$ and the observed incidence are available;

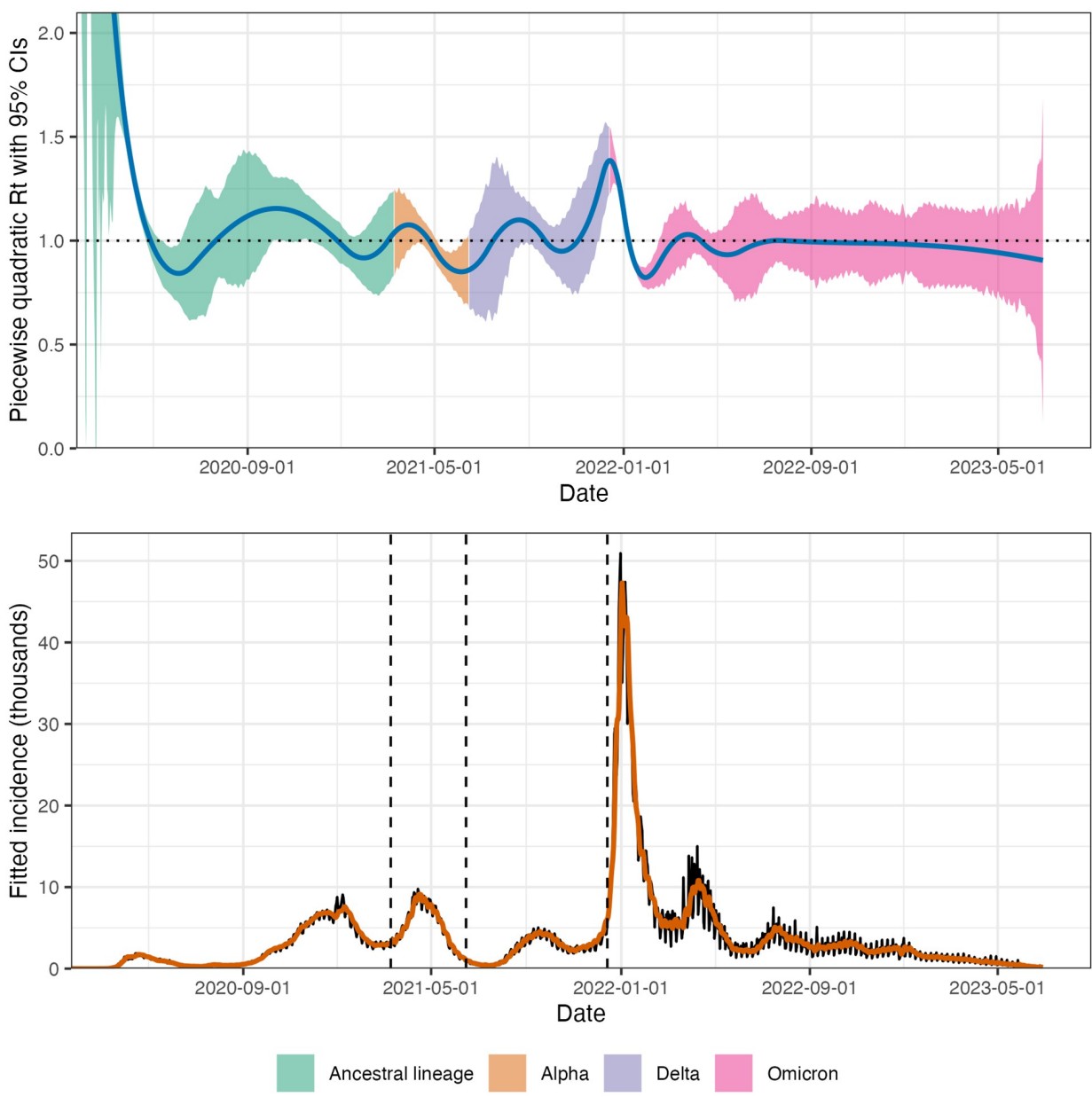

**Fig 1. A demonstration of instantaneous reproduction number estimation by `rtestim` and the corresponding predicted incident cases.**
The example is the Covid-19 epidemic in Canada during the period from January 23, 2020 to June 28, 2023. In the top panel, the blue curve is the estimated piecewise quadratic $\mathcal{R}_t$ and the colorful ribbon is the corresponding 95% confidence band. The colors represent the variants whose serial interval distributions are used to estimate $\mathcal{R}_t$. In the bottom panel, the black curve is the observed Covid-19 daily confirmed cases, and the orange curve on top of it is the predicted incident cases corresponding to the estimated $\mathcal{R}_t$. The three vertical dashed lines represent the beginning of a new dominant variant.

- the estimated log $\mathcal{R}_t$ can be mathematically described as an element of a well-known function space depending on user choice [28].

We use a proximal Newton method to solve the convex optimization problem along with warm starts to produce estimates efficiently, typically in a matter of seconds, even for long sequences of data. In a number of simulation experiments, we show empirically that our

approach is more accurate than existing methods at estimating the true instantaneous reproduction numbers and robust to some degrees of misspecification of incidence distribution, serial interval distribution, and the order of graphical curvature.

The manuscript proceeds as follows. We first introduce our $\mathcal{R}_t$ estimation methodology including the renewal equation and the development of Poisson trend filtering estimator. We explain how this method could be interpreted from the Bayesian perspective, connecting it to previous work in this context. We provide illustrative experiments comparing our estimator to other Bayesian alternatives. We then apply our methodology to the Covid-19 pandemic in Canada and the 1918 influenza pandemic in the United States. Finally, we conclude with a discussion of the advantages and limitations of our approach and describe some practical considerations for instantaneous reproduction number estimation.

## 2 Methods

### 2.1 Renewal model for incidence data

The instantaneous reproduction number $\mathcal{R}(t)$ is defined to be the expected number of secondary infections at time $t$ produced by a primary infection sometime in the past. To make this precise, denote the number of new infections at time $t$ as $y(t)$. Then the total primary infectiousness can be written as $\eta(t) := \int_0^\infty p(t, i)y(t - i)\mathrm{d}i$, where $p(t, i)$ is the probability that a new secondary infection at time $t$ is the result of a primary infection that occurred $i$ time units in the past. The instantaneous reproduction number is then given as the value that equates

$$\mathbb{E}[y(t) \mid y(j), \ j < t] = \mathcal{R}(t)\eta(t) = \mathcal{R}(t) \int_0^\infty p(t, i)y(t - i)\mathrm{d}i, \tag{1}$$

otherwise known as the renewal equation. The period between primary and secondary infections is exactly the generation time of the disease, but given real data, observed at discrete times (say, daily), this delay distribution must be discretized into contiguous time intervals, say, $(0, 1], (1, 2], \ldots$, resulting in the sequence $\{p_{t,i}\}_{i=0}^\infty$ corresponding to observations $y_t$ for each $t$ and yields the discretized version of Eq (1),

$$\mathbb{E}[y_t \mid y_1, \ldots, y_{t-1}] = \mathcal{R}_t\eta_t = \mathcal{R}_t \sum_{i=1}^\infty p_{t,i}y_{t-i}. \tag{2}$$

Many approaches to estimating $\mathcal{R}_t$ rely on Eq (2) as motivation for their procedures, among them, `EpiEstim` and `EpiFilter` [16].

In most cases, it is safe to assume that infectiousness disappears beyond $\tau$ timepoints ($p(t, i)$ = 0 for $i > \tau$), resulting in the truncated integral of the generation interval distribution $\int_0^\tau p(t, i)\mathrm{d}i = 1$ for each $t$. Generation time, however, is usually unobservable and tricky to estimate, so common practice is to approximate it by the serial interval: the period between the symptom onsets of primary and secondary infections. If the infectiousness profile after symptom onset is independent of the incubation period (the period from the time of infection to the time of symptom onset), then this approximation is justifiable: the serial interval distribution and the generation interval distribution share the same mean. However, other properties may not be similarly shared, and, in general, the generation interval distribution is a convolution of the serial interval distribution with the distribution of the difference between independent draws from the delay distribution from infection to symptom onset. See, for example, [3] for a fuller discussion of the dangers of this approximation. Nonetheless, treating these as interchangeable is common [10, 29], and doing otherwise is beyond the scope of this work. We will allow the delay distribution to be either constant over time—the probability $p(i)$

depends only on the gap between primary and secondary infections and not on the time $t$ when the secondary infection occurs—or to be time-varying: $p(t, i)$ also depends on the time of the secondary infection. For our methods, we assume that the serial interval can be accurately estimated from auxiliary data (say by contact tracing, or previous epidemics) and we take it as fixed, as is common in existing studies, [10, 19, 20].

The renewal equation in Eq (2) relates observable data streams (incident cases) occurring at different timepoints to the instantaneous reproduction number given the serial interval. The fact that it depends only on the observed incident counts makes it reasonable to estimate $\mathcal{R}_t$. However, data collection idiosyncrasies can obscure this relationship. Diagnostic testing targets symptomatic individuals, omitting asymptomatic primary infections which can lead to future secondary infections. Testing practices, availability, and uptake can vary across space and time [30, 31]. Finally, incident cases as reported to public health are subject to delays due to laboratory confirmation, test turnaround times, and eventual submission to public health [32]. For these reasons, reported cases are lagging indicators of the course of the pandemic. Furthermore, they do not represent the actual number of new infections that occur on a given day, as indicated by exposure to the pathogen. The assumptions described above (homogeneous mixing, similar susceptibility and social behaviours, etc.) are therefore consequential. That said, Eq (2) also provides some comfort about deviations from these assumptions. Under certain conditions, failing to account for the reporting behaviours will minimally impact the accuracy of any $\mathcal{R}_t$ estimator that is based on Eq (2). We discuss three types of deviation here. First, if $y_t$ is scaled by a constant $a$ describing the reporting ratio, then, because it appears on both sides of Eq (2), $\mathcal{R}_t$ will be unchanged. Second, if such a scaling $a_t$ varies in time, as long as it varies slowly relative to $p_i$ —that is, if $a_t / \sum_{i=1}^{t} a_i p_i \approx 1$—then $\mathcal{R}_t$ can still be estimated well from reported incidence data. Finally, if a sudden change in reporting ratio occurs at time $t_1$, it would only result in large errors in $\mathcal{R}_t$ at times near $t_1$ (where the size of this neighbourhood is determined indirectly by the effective support of $\{p_{t1}, i\}$). On the other hand, time-varying reporting delays would be much more detrimental [33, 34].

## 2.2 Poisson trend filtering estimator

We use the daily confirmed incident cases $y_t$ on day $t$ to estimate the observed infectious cases under the model that $y_t$, given previous incident cases $y_{t-1}, \ldots, y_1$ and a constant serial interval distribution, follows a Poisson distribution with mean $\Lambda_t$. That is,

$$y_t \mid y_1, \ldots, y_{t-1} \sim \text{Poisson}(\Lambda_t), \text{ where } \Lambda_t = \mathcal{R}_t \sum_{i=1}^{t-1} p_i y_{t-i} = \mathcal{R}_t \eta_t. \tag{3}$$

We will write $p_i$ as constant in time for simplicity, although this is not required. Given a history of $n$ confirmed incident counts $\mathbf{y} = (y_1, \ldots, y_n)^\top$, our goal is to estimate $\mathcal{R}_t$ for each $t = 1, \ldots, n$. A natural approach is to maximize the likelihood, producing the maximum likelihood estimator (MLE):

$$
\begin{aligned}
\widehat{\mathcal{R}} &= \underset{\mathcal{R} \in \mathbb{R}_+^n}{\operatorname{argmax}} \ \mathbb{P}(\mathcal{R} \mid \mathbf{y}, \ \mathbf{p}) = \underset{\mathcal{R} \in \mathbb{R}_+^n}{\operatorname{argmax}} \prod_{t=1,\ldots,n} \frac{(\mathcal{R}_t \eta_t)^{y_t} \ \exp\{-\mathcal{R}_t \eta_t\}}{y_t!} \\
&= \underset{\mathcal{R} \in \mathbb{R}_+^n}{\operatorname{argmin}} \sum_{t=1}^{n} \mathcal{R}_t \eta_t - y_t \log(\mathcal{R}_t \eta_t).
\end{aligned}
\tag{4}
$$

This optimization problem, however, is easily seen to yield a one-to-one correspondence between the observation and the estimated instantaneous reproduction number, i.e., $\widehat{\mathcal{R}}_t = y_t/\eta_t$, so that the estimated sequence $\widehat{\mathcal{R}}$ will have no significant smoothness.

The MLE is an unbiased estimator of the true parameter $\mathcal{R}_t$, but unfortunately has high variance: changes in $y_t$ result in proportional changes in $\widehat{\mathcal{R}}_t$. To avoid this behaviour, and to match the intuition that $\mathcal{R}_t \approx \mathcal{R}_{t-1}$, we advocate enforcing smoothness of the instantaneous reproduction numbers. This constraint will decrease the estimation variance, and hopefully lead to more accurate estimation of $\mathcal{R}$, as long as the smoothness assumption is reasonable. Smoothness assumptions are common (see e.g., [3, 16]), but the type of smoothness assumption is critical. Cori et al. [10] imposes smoothness indirectly by estimating $\mathcal{R}_t$ with moving windows of past observations. The Kalman filter procedure of [16] would enforce $\ell_2$-smoothness ($\int_0^n (\widehat{\mathcal{R}}''(t))^2 \mathrm{d}t < C$ for some constant $C$), although the computational implementation results in $\widehat{\mathcal{R}}$ taking values over a discrete grid. Pascal et al. [20] produces piecewise linear $\widehat{\mathcal{R}}_t$, which turns out to be closely related to a special case of our methodology. Smoother estimated curves will provide high-level information about the entire epidemic, obscuring small local changes in $\mathcal{R}(t)$, but may also remove the ability to detect large sudden changes, such as those resulting from lockdowns or other major containment policies.

To enforce smoothness of $\widehat{\mathcal{R}}_t$, we add a trend filtering penalty [28, 35–37] to Eq (5). Because $\mathcal{R}_t > 0$, we explicitly penalize the divided differences (discrete derivatives) of neighbouring values of $\log(\mathcal{R}_t)$. Let $\theta := \log(\mathcal{R}) \in \mathbb{R}^n$, so that $\Lambda_t = \eta_t \exp(\theta_t)$, and $\log(\eta_t \mathcal{R}_t) = \log(\eta_t) + \theta_t$. For evenly spaced incidence data, we write our estimator as the solution to the optimization problem

$$\widehat{\mathcal{R}} = \exp(\widehat{\theta}) \quad \text{where} \quad \widehat{\theta} = \underset{\theta \in \mathbb{R}^n}{\operatorname{argmin}} \ \eta^\mathsf{T} \exp(\theta) - \mathbf{y}^\mathsf{T}\theta + \lambda \| D^{(k+1)}\theta \|_1, \tag{5}$$

where $\exp(\cdot)$ applies elementwise and $\| \boldsymbol{a} \|_1 := \sum_{i=1}^n |a_i|$ is the $\ell_1$ norm. Here, $D^{(k+1)} \in \mathbb{Z}^{(n-k-1)\times n}$ is the $(k+1)^{\text{th}}$ order divided difference matrix for any $k \in \{0, \ldots, n-1\}$ with the convention that $D^{(0)} = \mathbf{0}_{n\times n}$. The divided difference matrix for $k = 0$, $D^{(1)} \in \{-1, 0, 1\}^{(n-1)\times n}$, is a sparse matrix with diagonal band of the form:

$$D^{(1)} = \begin{pmatrix} -1 & 1 & & & \\ & -1 & 1 & & \\ & & \ddots & \ddots & \\ & & & -1 & 1 \end{pmatrix}. \tag{6}$$

For $k \geq 1$, $D^{(k+1)}$ can be defined recursively as $D^{(k+1)} := D^{(1)}D^{(k)}$, where $D^{(1)} \in \{-1, 0, 1\}^{(n-k-1)\times(n-k)}$ has the form Eq (6) but with modified dimensions.

The tuning parameter (hyperparameter) $\lambda$ balances data fidelity with desired smoothness. When $\lambda = 0$, the problem in Eq (5) reduces to the MLE in Eq (4). Larger tuning parameters privilege the regularization term and yield smoother estimates. Finally, there exists $\lambda_{\max}$ such that any $\lambda \geq \lambda_{\max}$ will result in $D^{(k+1)}\widehat{\theta} = 0$ and $\widehat{\theta}$ will be the Kullback-Leibler projection of $\mathbf{y}$ onto the null space of $D^{(k+1)}$ (see Section 2.3 for more details).

The solution to Eq (5) will result in piecewise polynomials, specifically called discrete splines. For example, $0^{\text{th}}$-degree discrete splines are piecewise constant, $1^{\text{st}}$-degree curves are piecewise linear, and $2^{\text{nd}}$-degree curves are piecewise quadratic. For $k \geq 1$, $k^{\text{th}}$-degree discrete splines are continuous and have continuous discrete differences up to degree $k - 1$ at the knots

(i.e., changing points between segments). This penalty results in more flexibility compared to the homogeneous smoothness that is created by the squared $\ell_2$ norm. Using different orders of the divided differences results in estimated instantaneous reproduction numbers with different smoothness properties.

For unevenly spaced data, the spacing between neighbouring parameters varies with the time between observations, and thus, the divided differences must be adjusted by the times that the observations occur. Given observation times $\mathbf{x} = (x_1, \ldots, x_n)^\top$, for $k \geq 1$, define a $k^{\text{th}}$-order diagonal matrix

$$X^{(k)} = \text{diag}\left( \frac{k}{x_{k+1} - x_1}, \quad \frac{k}{x_{k+2} - x_2}, \quad \cdots, \quad \frac{k}{x_n - x_{n-k}} \right).\qquad(7)$$

Letting $D^{(\mathbf{x},1)} := D^{(1)}$, then for $k \geq 1$, the $(k+1)^{\text{th}}$-order divided difference matrix for unevenly spaced data can be created recursively by $D^{(\mathbf{x},\,k+1)} := D^{(1)} X^{(k)} D^{(\mathbf{x},\,k)}$. No adjustment is required for $k = 0$.

Due to the penalty structure, this estimator is locally adaptive, meaning that it can potentially capture local changes such as the initiation of control measures, becoming more wiggly in regions that require it. In contrast, Abry et al. and Pascal et al. considered only the $2^{\text{nd}}$-order ($k = 1$) divided difference of $\mathcal{R}_t$ rather than its logarithm [19, 20]. In comparison to their work, our estimator (i) allows for arbitrary degrees of temporal smoothness and (ii) avoids the potential numerical issues of penalizing/estimating positive real values. Nonetheless, as we will describe below, our procedure is computationally efficient for estimation over an entire sequence of hyperparameters $\lambda$ and provides methods for choosing how smooth the final estimate should be.

## 2.3 Solving over a sequence of tuning parameters

We can solve the Poisson trend filtering estimator over an arbitrary sequence of $\lambda$ that produces different levels of smoothness in the estimated curves. We consider a candidate set of M $\lambda$-values, $\boldsymbol{\lambda} = \{\lambda_m\}_{m=1}^M$, that is strictly decreasing.

Let $D := D^{(k+1)}$ for simplicity in the remainder of this section. As $\lambda \to \infty$, the penalty term $\lambda \|D\theta\|_1$ dominates the Poisson loss, so that minimizing Eq (5) is asymptotically equivalent to minimizing the penalty term, which results in $\|D\theta\|_1 = 0$. In this case, the divided differences of $\theta$ with order $k + 1$ is always 0, and thus, $\theta$ must lie in the null space of $D$, that is, $\theta \in \mathcal{N}(D)$. The same happens for any $\lambda$ beyond this threshold, so define $\lambda_{\max}$ to be the smallest $\lambda$ that produces $\theta \in \mathcal{N}(D)$. It turns out that this value can be written explicitly as $\lambda_{\max} = \| (D^\dagger)^\top (\eta - y) \|_\infty$, where $D^\dagger$ is the (left) generalized inverse of $D$ satisfying $D^\dagger D = I$ and $\|a\|_\infty := \max_i |a_i|$ is the infinity norm. Explicitly, for any $\lambda \geq \lambda_{\max}$, the solution to Eq (5) will be identical to the solution with $\lambda_{\max}$. Therefore, we use $\lambda_1 = \lambda_{\max}$ and choose the minimum $\lambda_M$ to be $r\lambda_{\max}$ for some $r \in (0, 1)$ (typically $r = 10^{-4}$). Given any $M \geq 3$, we generate a sequence of $\lambda$ values to be equally spaced on the log-scale between $\lambda_1$ and $\lambda_M$.

To compute the sequence of solutions efficiently, the model is estimated sequentially by visiting each $\lambda_m$ in order, from largest to smallest. The estimates produced for a larger $\lambda$ are used as the initial values (warm starts) for the next smaller $\lambda$. By solving through the entire sequence of tuning parameters, we improve computational efficiency and also enable one to trade between bias and variance, resulting in improved accuracy relative to procedures using a single fixed tuning parameter.

## 2.4 Choosing a final $\lambda$

We estimate model accuracy over the candidate set through $V$-fold cross validation (CV) to choose the best tuning parameter. Specifically, we divide $\mathbf{y}$ (except the first and last observations) roughly evenly and randomly into $V$ folds, estimate $\mathcal{R}_t$ for all $\lambda$ leaving one fold out, and then predict the held-out observations. Alternatively, one could use regular splitting, assigning every $v^{\text{th}}$ observation into the same fold. Note that our approach is most closely related to non-parametric regression rather than time series forecasting. That said, under some conditions, one can guarantee that $V$-fold remains valid for risk estimation in time series. The sufficient conditions are quite strong, but the guarantees are also stronger than would be required for model selection consistency [38].

Model accuracy can be measured by multiple metrics such as mean squared error $\text{MSE}(\widehat{y}, \ y) = n^{-1} \parallel \widehat{y} - y \parallel_2^2$ or mean absolute error $\text{MAE}(\widehat{y}, \ y) = n^{-1} \parallel \widehat{y} - y \parallel_1$, but we prefer to use the (average) deviance, to mimic the likelihood in Eq (4): $D(y, \ \widehat{y}) = n^{-1} \sum_{i=1}^n 2(y_i \log(y_i) - y_i \log(\widehat{y}_i) - y_i + \widehat{y}_i)$, with the convention that $0 \log(0) = 0$. Note that for any $V$ and any $M$, we will end up estimating the model $(V+1)M$ times rather than once.

## 2.5 Approximate confidence bands

We also provide empirical confidence bands of the estimators with approximate coverage. Consider the related estimator $\tilde{\mathcal{R}}_t$ defined as

$$\tilde{\mathcal{R}} = \exp(\tilde{\theta}) \quad \text{where} \quad \tilde{\theta} = \underset{\theta \in \mathbb{R}^n}{\operatorname{argmin}} \ \eta^{\mathsf{T}} \exp(\theta) - \mathbf{y}^{\mathsf{T}} \theta + \lambda \parallel D\theta \parallel_2^2 . \tag{8}$$

Letting $\tilde{\mathbf{y}} = \eta \circ \tilde{\mathcal{R}}$ (where $\circ$ denotes the elementwise product), it can be shown (for example, Theorem 2 in [39]) that an estimator for $\text{Var}(\tilde{\mathbf{y}})$ is given by $(\text{diag}(\tilde{\mathbf{y}}^{-2}) + \lambda D^{\mathsf{T}} D)^{\dagger}$. Finally, an application of the delta method shows that $\text{Var}(\tilde{\mathbf{y}}_t)/\eta_t^2$ is an estimator for $\text{Var}(\tilde{\mathcal{R}}_t)$ for each $t = 1, \ldots, n$. We therefore use $(\text{diag}(\widehat{\mathbf{y}}^{-2}) + \lambda D^{\mathsf{T}} D)_t^{\dagger}/\eta_t^2$ as an estimator for $\text{Var}(\widehat{\mathcal{R}}_t)$. An approximate $(1 - \alpha)\%$ confidence interval then can be written as $\widehat{\mathcal{R}}_t \pm s_t \times T_{\alpha/2, n-\text{df}}$, where $s_t$ is the square-root of $\text{Var}(\widehat{\mathcal{R}}_t)$ for each $t = 1, \cdots, n$ and df is the number of changepoints in $\widehat{\theta}$ plus $k + 1$ [36]. An approximate confidence interval of $\widehat{\mathbf{y}}$ can be computed similarly.

## 2.6 Bayesian perspective

Unlike many other methods for $\mathcal{R}_t$ estimation, our approach is frequentist rather than Bayesian. Nonetheless, it has a corresponding Bayesian interpretation: as a state-space model with Poisson observational noise, autoregressive transition equation of degree $k \geq 0$, e.g., $\theta_{t+1} = 2\theta_t - \theta_{t-1} + \varepsilon_{t+1}$ for $k = 1$, and Laplace transition noise $\varepsilon_{t+1} \sim \text{Laplace}(0, 1/\lambda)$. Compared to EpiFilter [16], we share the same observational assumptions, but our approach has a different transition noise. EpiFilter estimates the posterior distribution of $\mathcal{R}_t$, and thus it can provide credible interval estimates as well. Our approach produces the maximum *a posteriori* estimate via an efficient convex optimization, obviating the need for MCMC sampling. But the associated confidence bands are created differently.

## 3 Results

Implementation of our approach is provided in the R package rtestim. All computational experiments are conducted on the Cedar cluster provided by the Digital Research Alliance of Canada with R 4.3.1. The R packages used for simulation and real-data application are

`EpiEstim 2.2-4` [40], `EpiLPS 1.2.0` [41], and `rtestim 0.0.4`. The `R` scripts for `EpiFilter` are used [42].

### 3.1 Synthetic experiments

**3.1.1 Design for the synthetic data.** We simulate four scenarios of the instantaneous reproduction number, intended to mimic different epidemics. The first two scenarios are rapidly controlled by intervention, where the $\mathcal{R}(t)$ consists of one discontinuity and two segments. Scenario 1 has constant $\mathcal{R}(t)$ before and after an intervention, while Scenario 2 grows exponentially, then decays. The other two scenarios are more complicated, where more waves are involved. Scenario 3 has four linear segments with three discontinuities, which reflect the effect of an intervention, resurgence to rapid transmission, and finally suppression of the epidemic. Scenario 4 involves sinusoidal waves throughout the epidemic. The first three scenarios and the last scenario are motivated by [16] and [17] respectively. We name the four scenarios as (1) piecewise constant, (2) piecewise exponential, (3) piecewise linear, and (4) periodic.

In all cases, the times of observation are regular, and epidemics are of length $n = 300$. Specifically, in Scenario 1, $\mathcal{R}_t = 2$ for $t \leq 120$ and 0.8 for $t > 120$. In Scenario 2, $\mathcal{R}_t$ increases and decreases exponentially with rates 0.01 for $t \leq 100$ and 0.005 for $t > 100$. In Scenario 3, $\mathcal{R}_t$ is piecewise linear with four discontinuous segments,

$$
\begin{aligned}
\mathcal{R}(t) = \quad & \left(2.5 - \frac{0.5}{74}(t-1)\right)\mathbf{1}_{[1,76)}(t) + \left(0.8 - \frac{0.2}{74}(t-76)\right)\mathbf{1}_{[76,151)}(t) \\
& + \left(1.7 + \frac{0.3}{74}(t-151)\right)\mathbf{1}_{[151,226)}(t) + \left(0.9 - \frac{0.4}{74}(t-226)\right)\mathbf{1}_{[226,300]}(t),
\end{aligned}
\tag{9}
$$

where $\mathbf{1}_A(t) = 1$, if $t \in A$, and $\mathbf{1}_A(t) = 0$ otherwise. In Scenario 4, $\mathcal{R}_t$ is realization of the continuous, periodic curve generated by the function

$$
\mathcal{R}(t) = 0.2((\sin(\pi t/12) + 1) + (2\sin(5\pi t/12) + 2) + (3\sin(5\pi t/6) + 3)),
\tag{10}
$$

evaluated at equally spaced points $t \in [0, 10]$. These $\mathcal{R}_t$ scenarios are illustrated in Fig 2. We compute the expected incidence $\Lambda_t$ using the renewal equation, and generate the incident infections from the Poisson distribution with mean $\mathbb{E}[y_t \mid y_s, \ s < t] = \Lambda_t$. To verify the performance of our model under violations of the model's distributional assumptions, we also generate incident cases using the negative binomial distribution with dispersion parameter $\rho = 5$. Here, the negative binomial is parameterized such that the mean is $\mathbb{E}[y_t \mid y_s, \ s < t] = \Lambda_t$ and the variance is $\mathrm{Var}[y_t \mid y_s, s < t] = \Lambda_t(1 + \Lambda_t/\rho)$ (following, for example, [17]). Because $(1 + \Lambda_t/\rho) > 1$ for $0 \leq \rho < \infty$, this parameterization results in overdispersion relative to the Poisson distribution, with smaller values of $\rho$ leading to greater overdispersion. For context on the observed dispersion of these synthetic experiments, Fig A.2.1 in S1 Appendix displays the ratio of the time-varying standard deviation to the mean.

We use serial interval (SI) distributions of measles (with mean 14.9 and standard deviation 3.9) in Hagelloch, Germany in 1861 [43] and SARS (with mean 8.4 and standard deviation 3.8) in Hong Kong in 2003 [44], inspired by [10], to generate synthetic epidemics. We initialize all epidemics with $y_1 = 2$ cases and generate for $t = 2, \ldots, 300$. The synthetic measles epidemics have smaller incident cases in general, and the SARS epidemics have larger incidence. Essentially, the smaller mean of the serial interval for SARS with a similar standard deviation leads to shorter expected delays between onsets of primary and secondary infections, resulting in faster growth of incidence within the same period of time. We also consider shorter flu epidemics with 50 timepoints with piecewise linear $\mathcal{R}_t$ (Scenario 3) considering both incidence

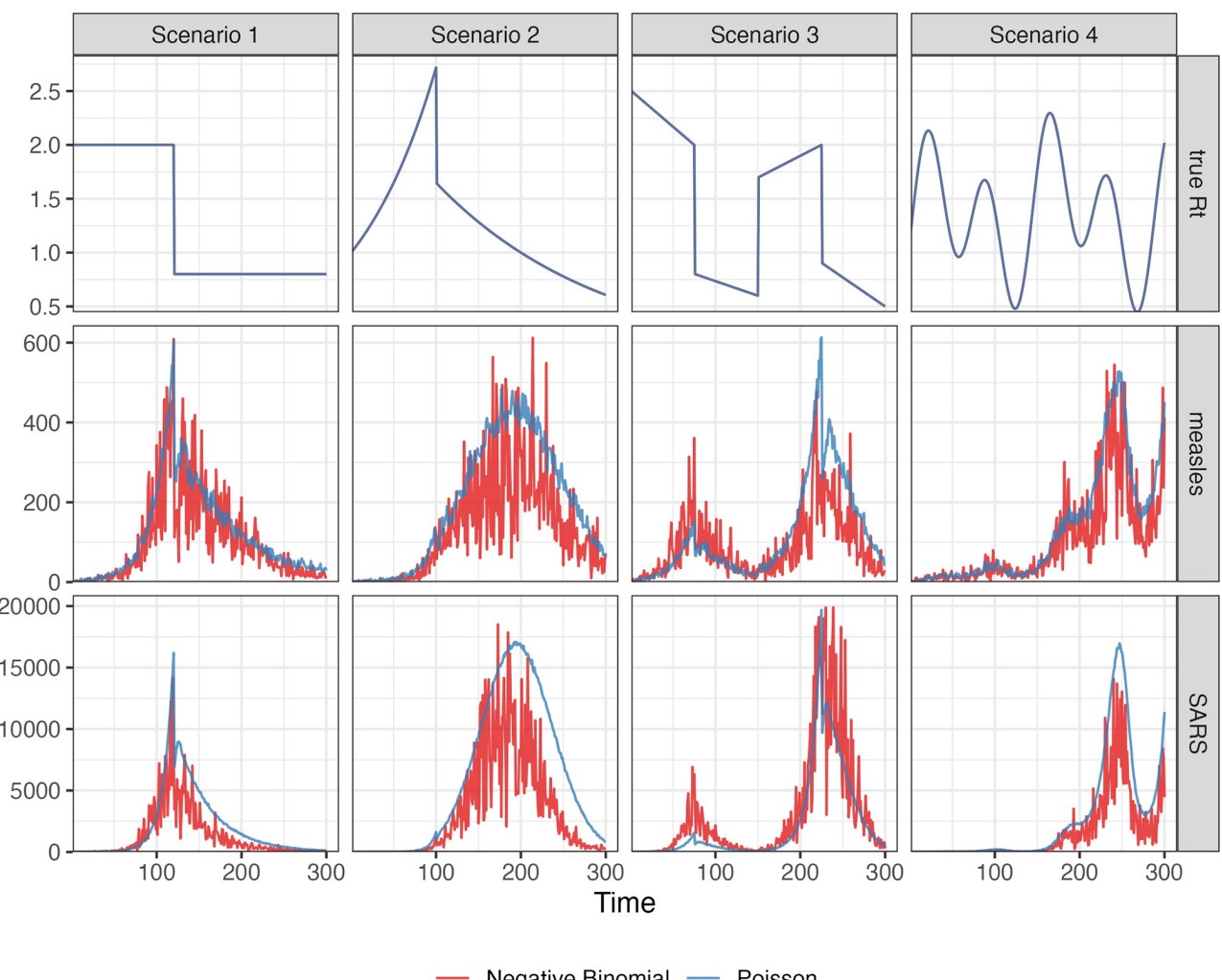

**Fig 2. This figure displays example realizations for each $\mathcal{R}_t$ setting.** Top row: the instantaneous reproduction numbers. Middle row: synthetic measles incidence (Poisson in blue, negative binomial in red) incidence. Bottom row: synthetic SARS incidence. The 4 $\mathcal{R}_t$ scenarios are shown in the columns.

distributional assumptions. The motivation is to compare our method and other alternatives with EpiNow2 which takes much longer to compute for long epidemics (nearly 2 hours to converge for a measles epidemic with 300 timepoints) than other methods. Besides using the correct SI distributions to estimate $\mathcal{R}_t$, we also consider the scenarios where the SI is misspecified. More details on experimental settings and results for shorter epidemics and misspecification of SI distributions are given in Sections A.2.1 and A.3 in S1 Appendix respectively.

For each problem setting (including SI distribution, an $\mathcal{R}_t$ scenario, and an incidence distribution), we generate 50 random samples, resulting in 800 total synthetic epidemics. Example realizations for measles and SARS with each instantaneous reproduction number scenario is displayed in Fig 2.

**3.1.2 Algorithmic choices.** We compare `rtestim` to `EpiEstim`, `EpiLPS`, and `EpiFilter`. `EpiEstim` estimates the posterior distribution of the instantaneous reproduction number given a Gamma prior and Poisson distributed observations over a trailing

window, under the assumption that the instantaneous reproduction number is constant during that window. A larger window averages out more fluctuations, leading to smoother estimates, whereas a shorter window is more responsive to sudden spikes or declines. We used weekly and monthly sliding windows, however, since neither considerably outperforms the other across all scenarios, we defer the monthly results to Section A.3.1 in S1 Appendix. `EpiLPS` is another Bayesian approach that estimates P-splines based on the Laplace approximation to the conditional posterior with negative binomial likelihood. It should more easily handle the negative binomial scenarios as it matches the data generating process. `EpiFilter` uses a particle filtering procedure on a discrete grid of possible $\mathcal{R}_t$ values.

In each setting, we apply `rtestim` with four choices of $k = 0, 1, 2, 3$ resulting in different shapes of the estimated $\mathcal{R}_t$—piecewise constant, piecewise linear, piecewise quadratic, and piecewise cubic—respectively. We use 10-fold cross validation (CV) to choose the parameter $\lambda$ that minimizes out-of-sample prediction risk from a candidate set of size 50, i.e., $\lambda = \{\lambda_1, \cdots, \lambda_{50}\}$, for long epidemics, and 5-fold CV for short epidemics (results for this case are deferred to Sections A.3.2 and A.4.2 in S1 Appendix). We select the tuning parameter that gives the lowest deviance between the estimated incidence and the held-out samples averaged over all folds.

For the alternative methods, we generally use the set of tuning parameters that were applied to their own experimental settings. We consider both weekly and monthly sliding windows in `EpiEstim`. `EpiLPS` uses 40 *P*-spline basis functions and optimizes using the Nelder-Mead procedure. For `EpiFilter`, we specify a grid with 2000 cells, use 0.1 for the size of the diffusion noise, and use the "smoothed" $\mathcal{R}_t$ (conditional on all data) as the final estimate.

For the $\mathcal{R}_t$ estimation using all models for each problem, we use the same serial interval distribution, that was used to generate synthetic data. Taking different hyperparameters into consideration, we solve each problem using 8 methods including `EpiEstim` with weekly or monthly sliding windows, `EpiLPS`, `EpiFilter`, and `rtestim` with piecewise constant, linear, quadratic, or cubic curves. We have not made any effort to tune these (and other choices) more carefully.

For `rtestim`, the choice of $k$ explicitly controls the function space to which the solution will belong [28], providing the analyst with a mathematical understanding of the result. When faced with real data, the choice of $k$ for `rtestim` should be done either (1) based on the analyst's preference for the resulting structure (e.g., "I want to find large jumps, so $k = 0$") or (2) in a data-driven manner, as a component of the estimation process. Our software enables both cases: the second case can be implemented by simply fitting different $k$ and choosing the set $k$, $\lambda$ that has smallest CV score. Thus, all necessary choices can be accomplished based solely on the data, a departure from existing methods in that we both allow this choice and provide simple data-driven methods to accomplish it.

**3.1.3 Accuracy measurement.** To measure estimation accuracy, we compare the estimated $\widehat{\mathcal{R}}$ to the true $\mathcal{R}$ using the Kullback-Leibler (KL) divergence. KL is useful in this context for a few reasons. First, it correctly handles the non-negativity constraint on $\mathcal{R}$. Second, KL matches the negative log-likelihood used in Eq (4). Third, it captures the curved geometry of the probability spaces implied by the Poisson distribution accurately. And fourth, as in the equation below, it has a convenient functional form depending only on $\mathcal{R}$ and $\eta$. For the Poisson distribution the KL divergence is defined as

$$D_{KL}(\mathcal{R} \parallel \widehat{\mathcal{R}}) = \sum_{t=1}^{n} \eta_t \left( \mathcal{R}_t \log\left(\frac{\mathcal{R}_t}{\widehat{\mathcal{R}}_t}\right) + \widehat{\mathcal{R}}_t - \mathcal{R}_t \right). \tag{11}$$

We use the average KL divergence: $\bar{D}_{KL}(\mathcal{R} \parallel \widehat{\mathcal{R}}) := D_{KL}(\mathcal{R} \parallel \widehat{\mathcal{R}})/n$. Details on the derivation of Eq (1) is provided in Section A.1 in S1 Appendix. KL divergence is more appropriate for measuring accuracy because it connects directly to the Poisson likelihood used to generate the data, whereas standard measures like the mean-squared error correspond to Gaussian likelihood. Using Poisson likelihood has the effect of increasing the relative cost of mistakes when $\Lambda_t$ is small.

To fairly compare across methods, we omit the first week of data (and estimates) for a few reasons. Estimates from EpiEstim are not available until $t = 8$ when using a weekly sliding window. Additionally, some procedures purposely impose strong priors that $\mathcal{R}_1$ is much larger than 1 to avoid over confidently asserting that an epidemic is under control. The effect of these priors will persist for days or weeks, but one would hope for accurate estimates as early in the outbreak as possible. Other details of the experimental settings are deferred to Section A.2 in S1 Appendix.

## 3.2 Results for synthetic data

In general, rtestim performs at least as well as the other competitors in the experimental study. Fig 3 visualizes the KL divergence across the seven methods with measles epidemics, and Fig 4 visualizes the counterparts with SARS epidemics. For low incidence in measles epidemics, rtestim is the most accurate for all $\mathcal{R}_t$ scenarios given both Poisson and negative binomial incidence. The best performance of rtestim has the lowest median and has little or no overlap with other methods. For Scenario 1 with Poisson incidence, EpiFilter has a similar median to that of rtestim and small variability, making it a competitive alternative. However for negative binomial incidence, EpiFilter loses its advantage and has the largest medians of any method in Scenarios 1 and 2. The large incidence in SARS epidemics is more difficult for all methods. For Poisson incidence, results are similar to the previous setting. However, for negative binomial incidence, EpiLPS performs at least as well if not better than rtestim, especially in Scenarios 2 and 4. Nonetheless, rtestim is largely similar, with simulation uncertainty suggesting comparable performance. We will examine a single realization of each experiment to investigate these global conclusions in more detail.

Fig 5 shows one realization for the estimated instantaneous reproduction number under the Poisson generative model in measles synthetic epidemics for all four scenarios. An expanded visualization with each estimated $\mathcal{R}_t$ curve displayed in a separate panel is provided in Fig A.6.1 in S1 Appendix. Ignoring the start of the epidemics, all methods look accurate and recover the underlying curves well, except EpiEstim with monthly sliding windows, where the trajectories are shifted to the right. Compared to EpiEstim and EpiLPS, which have rather severe difficulties at the beginning of the period, rtestim and EpiFilter estimates are more accurate without suffering from the initialization problem. The edge problem in EpiEstim and EpiLPS may be due to their priors, with the bias persisting for many days. A similar edge problem is possible for rtestim though it tends to be less severe for smaller $k$. Besides the edge problem, EpiEstim (especially, with the monthly sliding window) and EpiLPS produce "smooth" estimated curves that are continuous at the changepoints in Scenarios 1–3, resulting in large errors for a long period. Since the piecewise constant rtestim estimator does not enforce any smoothness, it easily captures the sharp change and nearly overlaps with the true values in Scenario 1. For larger $k$, rtestim can work nearly as well due to the $\ell_1$ penalty's ability to allow heterogenous smoothness. However, similar to other methods, rtestim has some difficulty with the first few timepoints, especially in the periodic scenario, where all methods fail to capture the first peak with much accuracy. EpiFilter recovers the $\mathcal{R}_t$ curves well in general, but tends to be more wiggly than other methods.

Fig 6 is similar to Fig 5 but shows estimated $\mathcal{R}_t$ given negative binomial incidence in SARS epidemics for each setting. An expanded visualization with each estimated $\mathcal{R}_t$ curve displayed

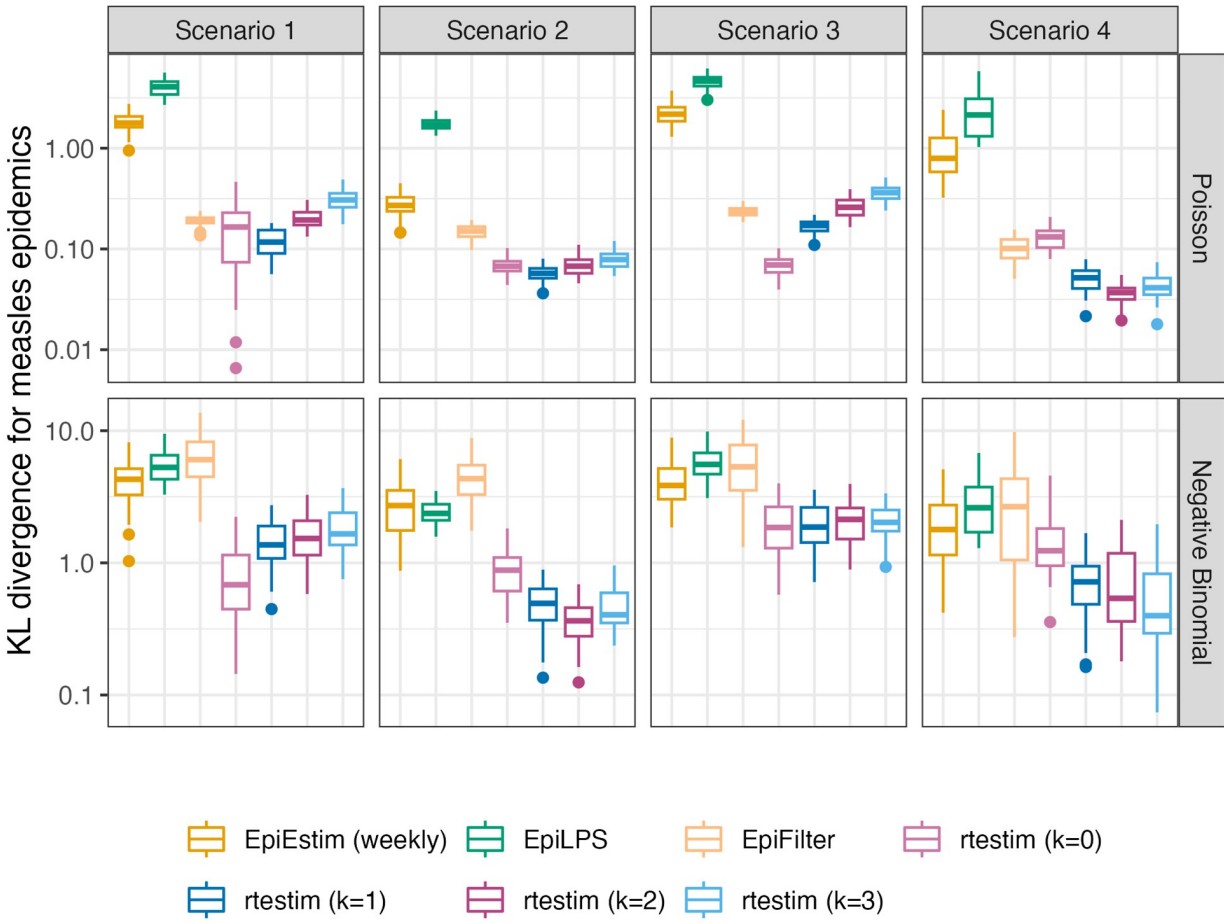

**Fig 3. Boxplot of mean KL divergence between $\widehat{\mathcal{R}}_t$ and true $\mathcal{R}_t$ across 50 synthetic measles epidemics.** Performance of each approach given Poisson incidence is in top panels and negative binomial incidence is in bottom panels. The average excludes the first week in all settings, since `EpiEstim` with a weekly sliding window does not provide estimates for the first week. Outliers beyond $1.5 \times$ IQR of each box are excluded for the sake of comparison with full range of the $y$-axis deferred to Fig A.3.1 in S1 Appendix.

in a separate panel is provided in Fig A.6.4 in S1 Appendix. Compared to the Fig 5, all methods perform worse overall for two main reasons: larger incidence and overdispersed data. All methods are worse at the start of the epidemics. `EpiFilter` is dramatically wiggly. The performance of `rtestim` is among the best in the first three $\mathcal{R}_t$ scenarios, though it has significant difficulties in the periodic scenario.

Finally, it is important to provide a brief comparison of the running times of all three models across the 8 experimental settings. We find that almost all models across all experiments operate within 10 seconds. Generally, `rtestim` takes the longest due to a relatively large number of estimates—50 values of $\lambda$ and 10 folds of cross validation require 550 estimates—while other models run only a single time for a fixed setting of their hyperparameters per experiment.

### 3.3 Real-data results: Covid-19 incident cases in Canada

We return to the data for Covid-19 confirmed incident cases in Canada examined in Section 1, where the dominant circulating variants are based on a multinomial logistic regression model

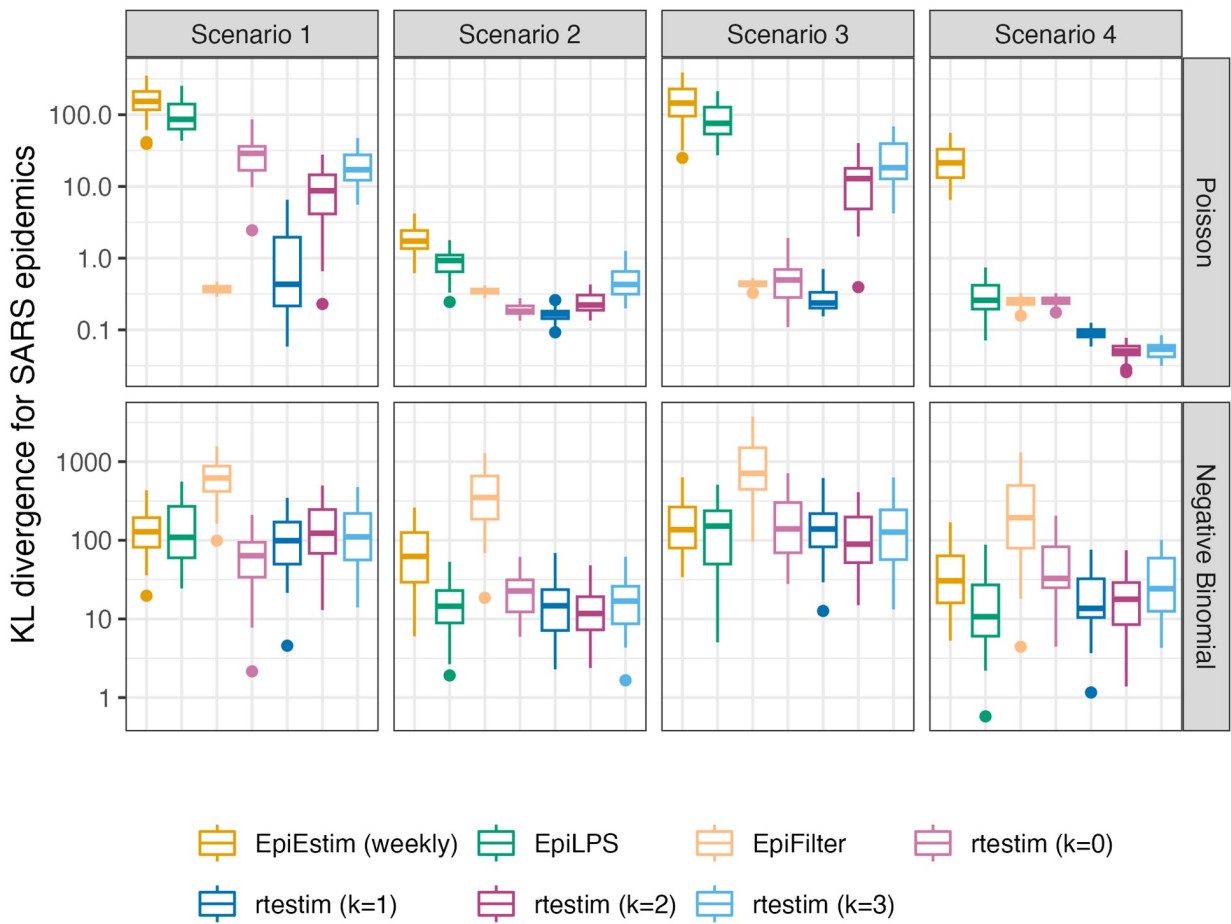

**Fig 4. Boxplot of mean KL divergence between $\widehat{\mathcal{R}}_t$ and true $\mathcal{R}_t$ across 50 synthetic SARS epidemics.** Performance of each approach given Poisson incidence is in top panels and negative binomial incidence is in bottom panels. The average excludes the first week in all settings, since EpiEstim with a weekly sliding window does not provide estimates for the first week. Outliers beyond $1.5 \times$ IQR of each box are excluded for the sake of comparison with full range of the $y$-axis deferred to Fig A.3.1 in S1 Appendix.

with variant probabilities from [45] and the corresponding serial interval distributions are based on results from [46]. Unlike in Fig 1, where we consider a time-varying serial interval distribution, in this section we use the weighted average of the serial interval distributions for the four dominant variants for the purposes of comparison with other methods, none of which allow time-varying delays. The estimates produced by rtestim are displayed in Fig 7 while the estimates of all competitors are deferred to Figs A.8.1 and A.8.2 in S1 Appendix.

Considering $k$ = 1, 2 and 3, $\widehat{\mathcal{R}}_t$ for Covid-19 in Canada is always less than 2 except at the very early stage, which means that one distinct infected individual on average infects less than two other individuals in the population. Examining three different settings for $k$, the temporal evolution of $\widehat{\mathcal{R}}$ (across all regularization levels $\lambda$) are similar near the highest peak around the end of 2021 before dropping shortly thereafter. Throughout the estimated curves, the peaks and troughs of the instantaneous reproduction numbers precede the growth and decay cycles of confirmed cases, as expected. We also visualize 95% confidence bands for the point estimates with $\lambda$ chosen by minimizing cross-validated KL divergence in Fig 7.

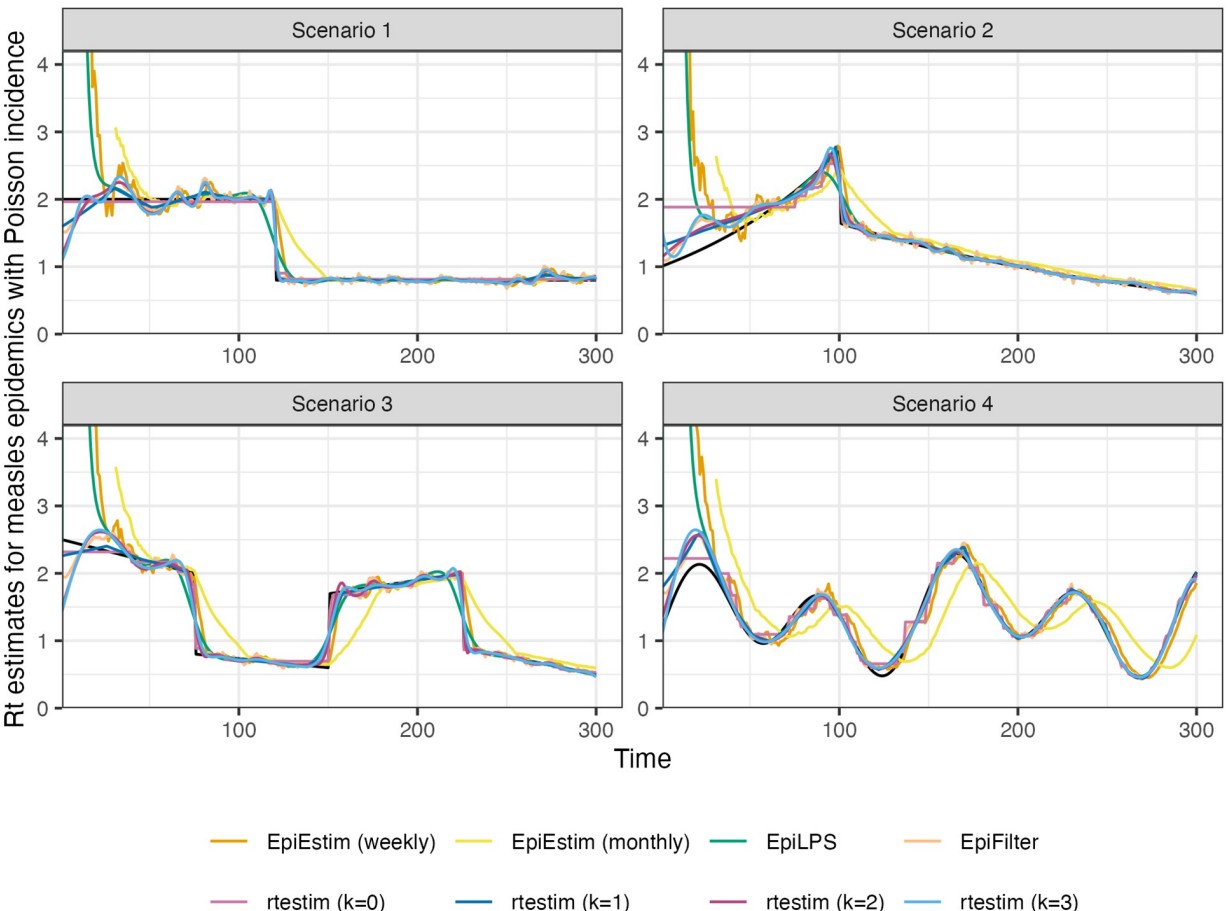

**Fig 5. $\mathcal{R}_t$ estimates for realizations of a measles epidemic with Poisson observations.** An expanded visualization with each estimated $\mathcal{R}_t$ curve displayed in a separate panel is provided in Fig A.6.1 in S1 Appendix.

The estimated instantaneous reproduction numbers are relatively unstable before April, 2022. The highest peak coincides with the emergence and global spread of the Omicron variant. The estimated instantaneous reproduction numbers fall below 1 during a few time periods, where the most obvious troughs are roughly from April 2021 to July 2021 and from January, 2022 to April 2022. The first trough coincides with the introduction of Covid-19 vaccines in Canada. The second trough, shortly after the largest peak may be due to variety of factors resulting in the depletion of the susceptible population such as increased self-isolation in response to media coverage of the peak or immunity incurred via recent infection. Since April 2022, the estimated instantaneous reproduction number has remained relatively stable (fluctuating around one) corresponding to low reported cases, though reporting behaviours also changed significantly since the Omicron wave.

## 3.4 Real-data results: Influenza in Baltimore, Maryland, 1918

We also apply rtestim to daily reported influenza cases in Baltimore, Maryland occurring during the world-wide pandemic of 1918 from September to November [47]. The data, shown in Fig 8, is included in the EpiEstim R package, along with the serial interval distribution. The 1918 influenza outbreak, caused by the H1N1 influenza A virus, was unprecedentedly

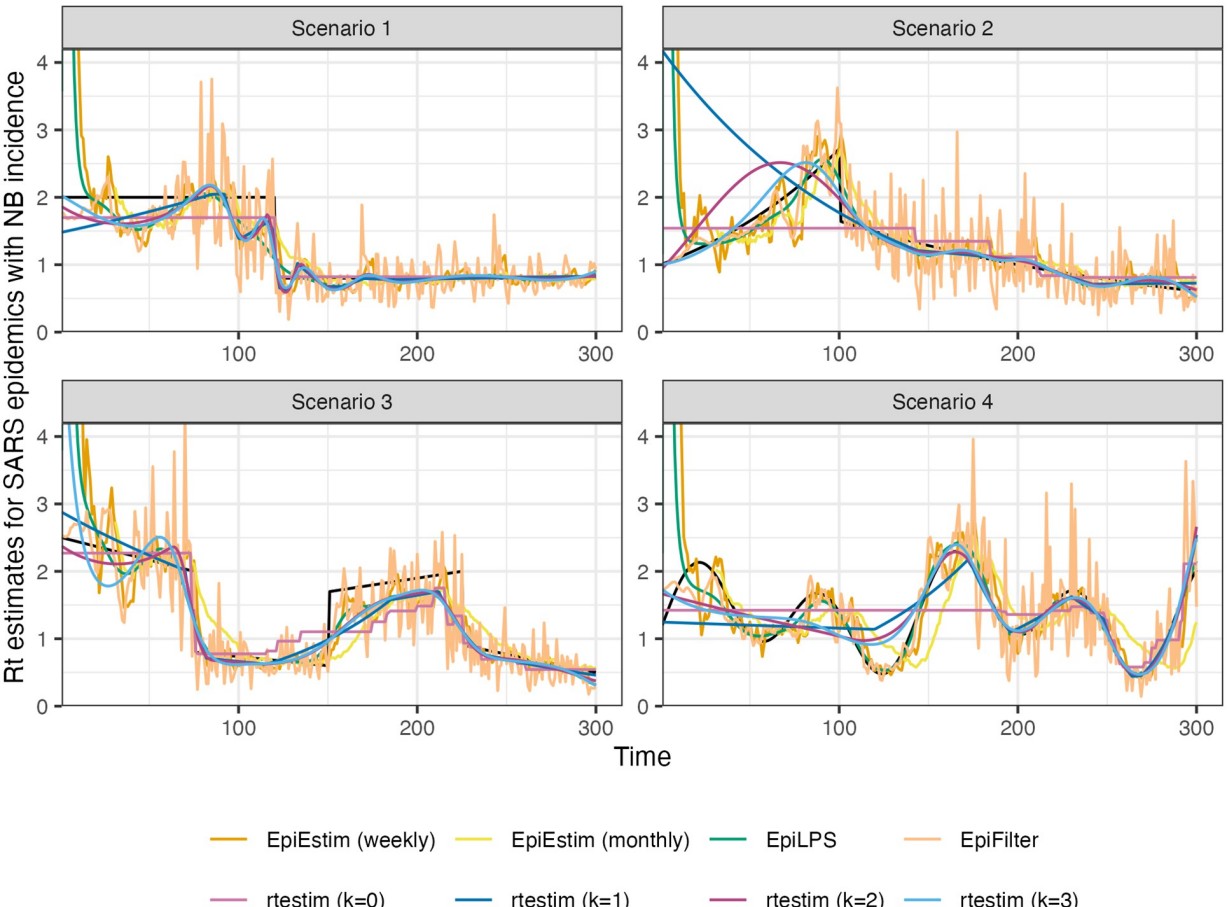

**Fig 6. $\mathcal{R}_t$ estimates for realizations of a SARS epidemic with negative binomial observations.** An expanded visualization with each estimated $\mathcal{R}_t$ curve displayed in a separate panel is provided in Fig A.6.4 in S1 Appendix.

deadly with case fatality rate over 2.5%, infecting almost one-third of the population across the world [48]. The CV-tuned piecewise cubic estimates in Fig 9 better capture the growth at the beginning of the pandemic in Fig 8. The estimated $\mathcal{R}_t$ curve suggests that the transmissibility of the pandemic grew rapidly over the first 30 days before declining below one after 50 days. However, it also suggests an increase in infectiousness toward the end of the period. With this data, it is difficult to determine if there is a second wave or a steady decline ahead. The CV-tuned piecewise constant and linear estimates in Fig 9 both suggest a steady decline. This conclusion is supported by the fact that incident cases decline to zero at the end of the period, matching $\mathcal{R}_t$ estimates in [10], which are all lower than one. Results from alternative software is deferred to Figs A.8.3 and A.8.4 in S1 Appendix.

## 4 Discussion

Our methodology provides a locally adaptive estimator using Poisson trend filtering. It captures the heterogeneous smoothness of instantaneous reproduction numbers given observed incidence data rather than resulting in global smoothness. This is a nonparametric regression model which can be written as a convex optimization problem. Minimizing the negative logliklihood of observations guarantees data fidelity while the penalty on divided differences

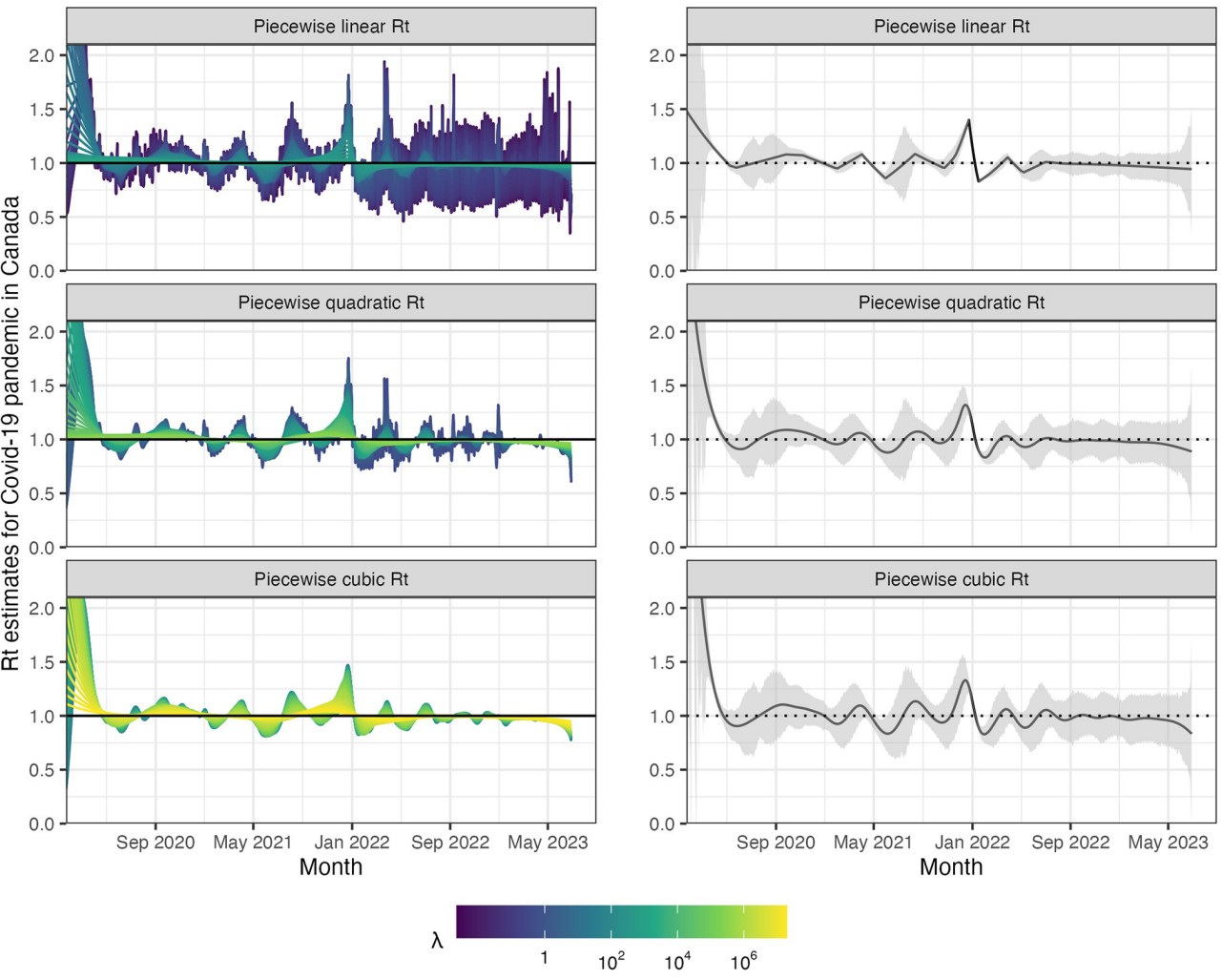

**Fig 7. Estimated instantaneous reproduction number based on Covid-19 daily confirmed incident cases.** The epidemic is between January 23rd, 2020 and June 28th, 2023 in Canada. The left panels show estimates corresponding to 50 tuning parameters. The right panels show the CV-tuned estimate along with approximate 95% confidence bands. The top, middle and bottom panels show the estimated $\mathcal{R}_t$ using the Poisson trend filtering in Eq (5) with degrees $k = 1, 2, 3$ respectively. All estimates use a constant serial interval distribution, which is the weighted sum of probabilities of the 4 dominant variants used in Fig 1.

between pairs of neighbouring parameters imposes smoothness. The $\ell_1$-regularization results in sparsity of the divided differences, leading to heterogeneous smoothness across time.

The property of local adaptivity (heterogenous smoothness) is useful to automatically distinguish, for example, seasonal outbreaks from outbreaks driven by other factors (behavioural changes, foreign introduction, etc.). Given a well-chosen polynomial degree, the growth rates can be quickly detected, potentially advising public health authorities to implement policy changes. The instantaneous reproduction numbers can be estimated retrospectively to examine the efficacy of such policies, whether they result in $\mathcal{R}_t$ falling below 1 or the speed of their effects. The smoothness of $\mathcal{R}_t$ curves (including the polynomial degrees and tuning parameters) should be chosen based on the purpose of the study in practice or with data-driven risk estimation by cross validation.

Our method provides a natural way to deal with missing data, for example, on weekends and holidays or due to changes in reporting frequency. While solving the convex optimization

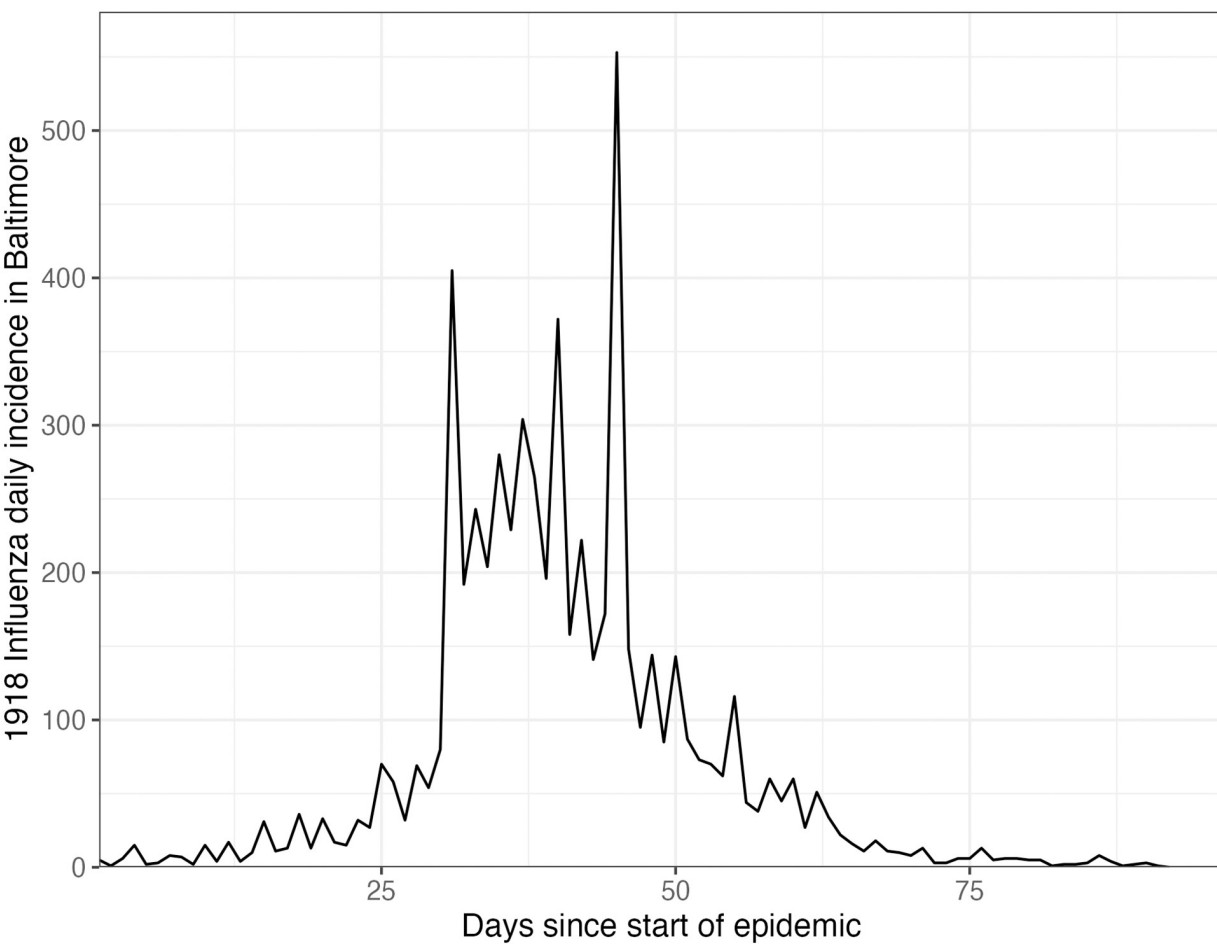

**Fig 8. Daily incident influenza cases in Baltimore, Maryland between September and November 1918.**

problem, our method can easily handle uneven spacing or irregular reporting. Computing the total primary infectiousness is also easily generalized to irregular reporting through automatic modifications of the discretization of the serial interval distribution. However, there are many other aspects to be considered in choosing the delay distribution to improve accuracy [34]. Additionally, because the $\ell_1$ penalty introduces sparsity (operating like a median rather than a mean), this procedure is relatively insensitive to spurious outliers compared to $\ell_2$ regularization.

There are a number of limitations that may influence the quality of $\mathcal{R}_t$ estimation. While our model is generic for incidence data rather than tailored to any specific disease, it does assume that the generation interval is short relative to the period of data collection. More specialized methodologies would be required for diseases with long incubation periods such as HIV or Hepatitis. Our approach, does not explicitly model imported cases, nor distinguish between subpopulations that may have different mixing behaviour. However, a natural extension to handle imported cases is to follow the suggested procedure of [11]. By including imported cases only in $\eta_t$ rather than in both $y_t$ and $\eta_t$, we exclude individuals who were infected elsewhere, lowering $\mathcal{R}_t$, but correctly reflecting the number of new primary infectees.

While the Poisson assumption is common, it does not handle overdispersion (observation variance larger than the mean). The negative binomial distribution is a good alternative, but

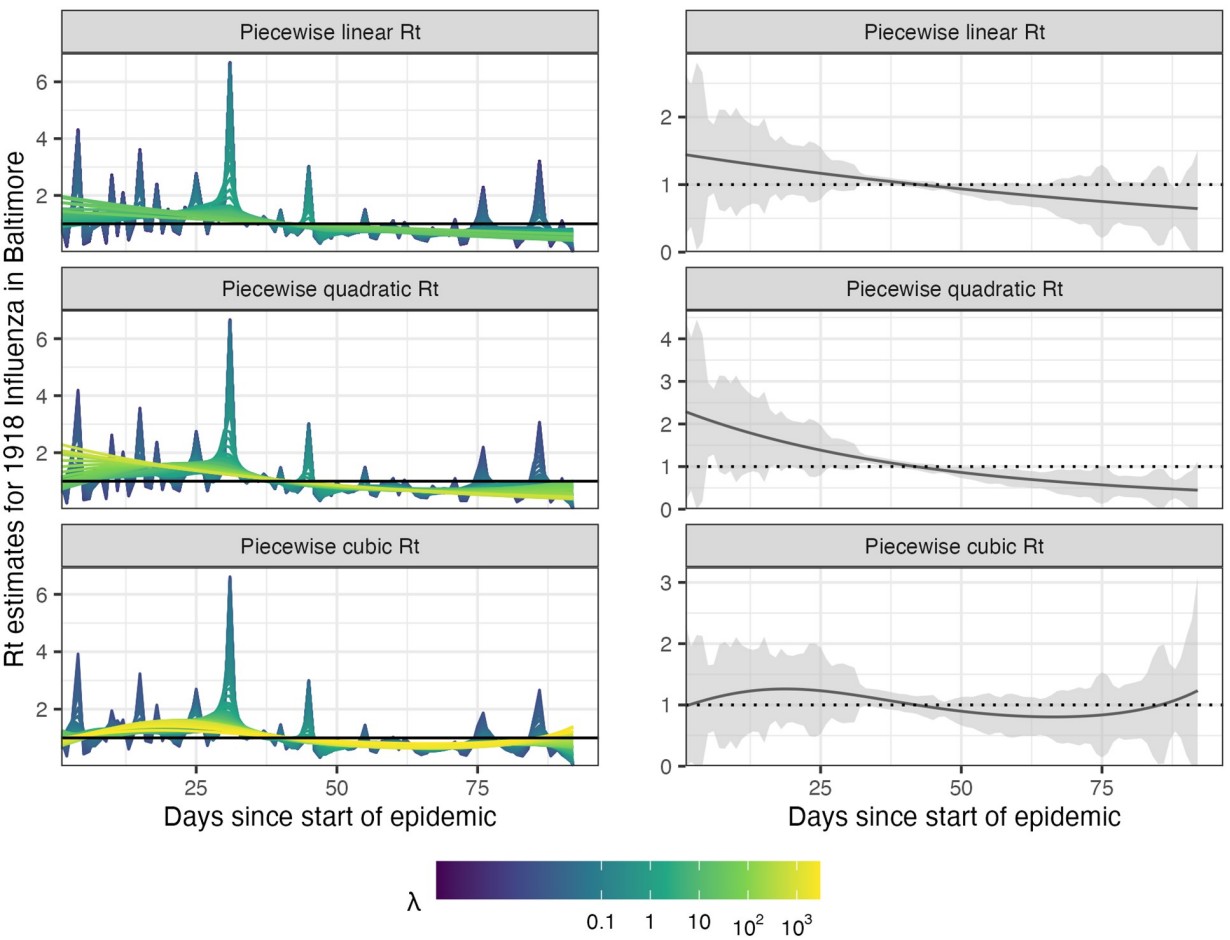

**Fig 9. Estimated instantaneous reproduction numbers for influenza in Baltimore, Maryland in 1918.** The left panels show estimates for a set of 50 tuning parameters. The right column displays the CV-tuned estimates with approximate 95% confidence bands. The rows (top to bottom) show $\widehat{\mathcal{R}}_t$ using Poisson trend filtering with $k = 1, 2, 3$ respectively.

more difficult to estimate in this context. As described in Section 1, the expression for $\mathcal{R}$ assumes that a relatively constant proportion of true infections is reported. However, if this proportion varies with time (say, due to changes in surveillance practices or testing recommendations), the estimates may be biased over this window. A good example is in early January 2022, during the height of the Omicron wave, Canada moved from testing all symptomatic individuals to testing only those in at-risk groups. The result was a sudden change that would render $\mathcal{R}_t$ estimates on either side of this timepoint incommensurable.

Our implementation in the software package rtestim can accommodate a fixed serial interval throughout the period of study (as implemented in simulation and in the real epidemics) or use time-varying serial interval distributions (as implemented in Fig 1 for Covid-19 data in Canada). In reality, the serial interval may vary due to changes in the factors such as population immunity [12]. One issue regarding the serial interval distribution relates to equating serial and generation intervals (also mentioned above). The serial interval distribution is generally wider than that of the generation interval, because the serial interval involves the convolution of two distributions, and is unlikely to actually follow a named distribution like gamma, though it may be reasonably well approximated by one. Our implementation allows

for an arbitrary distribution to be used, but requires the user to specify the discretization explicitly, requiring more nuanced knowledge than is typically available. Pushing this analysis further, to accommodate other types of incidence data (hospitalizations or deaths), a modified generation interval distribution would be necessary, and further assumptions would be required as well. Or else, one would first need to deconvolve deaths to infection onset before using our software.

Accurate statistical coverage of a function is a difficult problem, and the types of (frequentist) guarantees that can be made are not always what one would want [49]. We examine the coverage of our approximate confidence interval in simulation, with details are deferred to Section A.6 in S1 Appendix. Empirically, our observations for our method, as well as all others we have seen, follow a similar (undesirable) pattern: when $\mathcal{R}_t$ is stable, they over cover dramatically (even implausibly narrow intervals have 100% coverage); but when $\mathcal{R}_t$ changes abruptly, they under cover. Theoretically, whether these intervals should be expected to provide $(1 - \alpha)$% coverage simultaneously over all time while being narrow enough to provide useful uncertainty quantification is neither easy nor settled. An alternative to our approximation in Section 2.5, which we defer to future work, is to use the data fission method proposed by [50], which provides post-selection inference for trend filtering.

Nonetheless, our methodology is implemented in a lightweight R package rtestim and computed efficiently, especially for large-scale data, with a proximal Newton solver coded in C++. Given available incident case data, prespecified serial interval distribution, and a choice of degree $k$, rtestim is able to produce accurate estimates of instantaneous reproduction number and provide efficient tuning parameter selection via cross validation.

## Supporting information

**S1 Appendix. Supplement for "rtestim: Time-varying reproduction number estimation with trend filtering".** This includes eight sections, Sections A.1–A.8. Section A.1: Derivation of Kullback Leibler divergence. Section A.2: Additional details on experimental settings. Section A.3: Additional accuracy comparisons. Section A.4: Experimental results under misspecification of the serial interval distributions. Section A.5: Time comparisons of all methods. Section A.6: Confidence interval coverage. Section A.7: Data examples and alternative visualizations of Figs 5 and 6. Section A.8: Application of rtestim and all competitors on real epidemics.
(PDF)

## Acknowledgments

This research was enabled in part by support provided by BC DRI group who manages Cedar cloud (https://docs.alliancecan.ca/wiki/Cedar) and the Digital Research Alliance of Canada (alliancecan.ca).

## Author Contributions

**Conceptualization:** Paul Gustafson, Daniel J. McDonald.

**Formal analysis:** Jiaping Liu.

**Funding acquisition:** Paul Gustafson, Daniel J. McDonald.

**Investigation:** Jiaping Liu.

**Methodology:** Jiaping Liu, Zhenglun Cai, Paul Gustafson, Daniel J. McDonald.

**Resources:** Daniel J. McDonald.

**Software:** Jiaping Liu, Zhenglun Cai, Daniel J. McDonald.

**Supervision:** Daniel J. McDonald.

**Validation:** Jiaping Liu, Daniel J. McDonald.

**Visualization:** Jiaping Liu, Daniel J. McDonald.

**Writing – original draft:** Jiaping Liu.

**Writing – review & editing:** Jiaping Liu, Zhenglun Cai, Paul Gustafson, Daniel J. McDonald.

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
