## [Decision Letter · Decision Letter 0]

26 Feb 2024

Dear Ms Liu,

Thank you very much for submitting your manuscript "RtEstim: Effective reproduction number estimation with trend filtering" for consideration at PLOS Computational Biology.

As with all papers reviewed by the journal, your manuscript was reviewed by members of the editorial board and by several independent reviewers. In light of the reviews (below this email), we would like to invite the resubmission of a significantly-revised version that takes into account the reviewers' comments.

Reviewers' key concerns focused on the robustness of the proposed method. They noted that the choice of parameters k and λ, as well as the impact of Generation Interval misspecification, require further attention to ensure the method's reliability and generalizability.

We cannot make any decision about publication until we have seen the revised manuscript and your response to the reviewers' comments. Your revised manuscript is also likely to be sent to reviewers for further evaluation.

Sincerely,

Claudio José Struchiner, M.D., Sc.D.

Academic Editor

PLOS Computational Biology

Virginia Pitzer

Section Editor

PLOS Computational Biology

Reviewers' key concerns focused on the robustness of the proposed method. They noted that the choice of parameters k and λ, as well as the impact of Generation Interval misspecification, require further attention to ensure the method's reliability and generalizability.

Reviewer's Responses to Questions

**Comments to the Authors:**

Reviewer #1: The review is uploaded as an attachment.

Reviewer #2: In this manuscript the authors are presenting a new method for estimating retrospectively the time-dependent effective reproduction number by reducing the problem to an optimisation with an L1 penalty term to ensure smoothness of solution and Leave One Out cross-validation for fine-tuning. A comprehensive case study analysis is also done in the manuscript by testing it against two other methods (which at present are the go-to methods for fast inference of Rt), using both synthetic data and real case data.

The method is well presented in the context of the available literature on similar methods of fast inference of Rt trajectories and the authors do well in introducing the modelling framework in terms of the renewal equation. The analysis is sound in terms of mathematical accuracy and is fairly easy to follow along as it includes a detailed explanation for how the range of queried values in the hyperparameter space of the smoothness parameter \\lambda is computed.

The results shown in Figures 3 and 4 are very promising, though I am wondering how sensitive the proposed method is to the choice of smoothness degree k of the discrete splines. In the context of the synthetic datasets (both Figure 3 and 10 and 11) it seems that the choice of the parameter k plays an important role in whether the RtEstim method will outperform the other two considered, as it can be observed by comparing the means in Scenario 2. Also, the EpiLPS seems to generate relatively similar results in the case of Scenario 2 to those produced by RtEstim, at a fraction of the time.

Also, it is not very clear how well the proposed RtEstim method would behave in the context of a misspecified k parameter choice. This becomes really important in the real data analysis where we do not know a priori the type and number of temporal changes we can expect. A clearer explanation of the reason behind the choices of k made and how one would select an optimal k would be useful for completeness.

Two minor comments: A more accurate definition of the effective reproduction number Rt is that it is the expected number of secondary infections produced by primary infection throughout the course of his entire infection if conditions remain the same as at time t. Also, in Line 220 using the phrase “ M \\lambda-values” would help improve clarity of text.

Reviewer #3: Attachment uploaded.

Reviewer #4: The authors develop an approach for learning the effective reproduction number (Rt) from incidence data. In common with several other methods described at the beginning of the paper, they use the renewal equation for daily cases, with a Poisson distribution to describe stochasticity in the cases. The authors propose learning Rt via maximum likelihood, coupled with a penalty term on divided differences of log(Rt). Their penalty term is parameterized by two hyperparameters: a strength term lambda, controlling how strong the regularization is, and an order term k, such that the (k+1)th order divided differences are used in the regularization. When applying the method, the authors propose that the value of lambda be selected by running on a grid of possible lambda values and selecting the best via cross validation.

The manuscript is clearly written and deals with an important and interesting problem (how best to learn time variation in epidemiological parameters such as Rt).

Major concerns

p.13. EpiFilter would be an important comparator method here, as the authors have mentioned it several times earlier in their paper. However, they remove it from consideration after finding that it fails to converge for some of their synthetic data, due to the large incidence counts. However, I have actually been able to run EpiFilter to learn Rt successfully on incidence time series with larger numbers of daily cases than those considered in this paper. I appreciate that there may be some other features of the author’s time series which cause EpiFilter to fail, but if EpiFilter is going to be discussed as a comparator earlier in the paper and then not actually used, I would value a more detailed explanation of why it cannot be applied in the setting considered by the authors.

p.14. These challenges with EpiFilter draw attention to another concern, which is that in their synthetic data experiments, despite considering a range of different shapes of true Rt profiles, the authors are always relying on time series which reach high daily incidence (>1000/day). It would be valuable to see how the methods perform when incidence is much lower.

p.14. RtEstim computes Rt for a large grid of lambda values, with the best selected via cross validation. I acknowledge that tuning lambda is an integrated part of the RtEstim method, and the authors have developed an efficient way to perform this, whereas hyperparameter tuning may be more of a computational burden with the other methods. However, at the very least there should be more discussion of how the tuning parameters of EpiEstim and EpiLPS were selected, and whether those values are appropriate for the problems at hand, given the possibility of a situation where the comparators are only underperforming due to inappropriate hyperparameter choices.

p.17. EpiEstim and EpiLPS are both seeing drastic overestimation of Rt for the first few days of the time series. The authors mention this, but don’t explain why it’s happening. Is there something going wrong with the methods or their implementations to cause this? Are the results where RtEstim appears to outperform overall (Figure 3) arising just because the other methods have an overestimation error in the first few days?

p.23. If there is some substantial computational, theoretical, or implementation detail that makes imported cases difficult to incorporate into RtEstim, I am fine with the authors explaining this and deferring it to future work. But, would it actually be at all difficult to allow the RtEstim method and software to incorporate imported cases, using a modelling framework such as [2]? Many of the comparator methods and software packages discussed in the paper already have the ability to handle imported cases.

General. I would argue that an important part of learning Rt is estimating the uncertainty in Rt implied by the data. In many situatons, uncertainty in the value of Rt can be an important part of decision making. For example, if some measure of the best fit Rt for a particular incidence series falls below 1, but this is an uncertain estimate and there is still a decent chance that Rt is actually >1, it would be highly unwise to conclude that the disease outbreak is currently under control. The authors develop confidence bounds for their method in section 2.5, but they don’t make much use of them when they compare to other methods, nor do they discuss the implications of the confidence bounds when fitting to real data. Similarly, they discuss the robustness of their method and other methods to negative binomial stochasticity, but the focus again is on daily point estimates of Rt rather than considering to what extent the inferred uncertainty encompasses the true value of Rt.

Minor concerns

p.2, l.4. See [1] for “instantaneous” vs. “case” effective reproduction number.

p.3, l.23 et seq. Bayesian approaches to learning Rt also enable any prior knowledge about the value of the parameter to be incorporated into inference. Some studies (e.g., [2]) have specified priors on Rt with prior mean substantially above 1 to ensure conservative posterior estimates.

p.3, l.51. This limitation, while definitely a valid point to make, does not necessarily apply to methods using conjugate priors (as mentioned by the authors on p.3, l.29), some of which can be computed almost instantaneously, so I would adjust the wording here.

p.15. l.322. The derivation of the authors’ divergence measure could be elaborated.

p. 22, l.436. Discussion of the importance and role of the k parameter, and best practices for setting it, could be slightly elaborated, with reference made back to some of the results earlier in the paper.

[1] Gostic, Katelyn M., et al. "Practical considerations for measuring the effective reproductive number, Rt." PLOS Computational Biology 16.12 (2020).

[2] Thompson, Robin N., et al. "Improved inference of time-varying reproduction numbers during infectious disease outbreaks." Epidemics 29 (2019).

Reviewer #5: The authors have developed an approach to enable smoothing for the estimation of the effective reproduction number. The method is based on a trend filtering objective function with discrete splines and is solved by the proximal Newton method. Simulations were performed to evaluate the performance of the proposed approach compared to other methods. Real data analysis was conducted to demonstrate the value of implementing the proposed method in real-world scenarios. The paper is well-written and addresses important practical issues when estimating the effective reproduction number.

I have following minor suggestions/questions:

1. Line 102 - 104: the sentence seems to be missing a break between "... observed at discrete times (say, daily)" and "this delay distribution must be ...". Please rephrase if needed.

2. It's essential to choose the order k carefully for the smoothness of the estimator. Besides relying on domain knowledge and subjective preferences, are there more objective methods to determine the proper order? Are there any general suggestions for selecting k and assessing whether it's well-chosen?

3. In the simulations, why weren't all k = 0, 1, 3 evaluated in every scenario? While it's logical for k to correspond to certain R_t shapes, shouldn't we avoid basing the choice of k on synthetic R_t shapes, given that the true R_t wouldn't be known a priori in real-world data analysis? It would be insightful to assess the estimator's robustness by evaluating all k = 0, 1, 3 and showcasing the estimates with these k values in all scenarios.

4. Line 184, why the dimension of D^{(1)} still contain k?, shouldn't it only contain n?

5. While performing cross-validation to select \\lambda, are there any suggestions on determining the range of \\lambda to search over?

6. in line 433 - 434: For "... potentially advising public health to implement ...", should it be something like "... potentially advising public health authorities to implement ..."?

**Have the authors made all data and (if applicable) computational code underlying the findings in their manuscript fully available?**

Reviewer #1: Yes

Reviewer #2: Yes

Reviewer #3: Yes

Reviewer #4: Yes

Reviewer #5: Yes

PLOS authors have the option to publish the peer review history of their article (what does this mean?). If published, this will include your full peer review and any attached files.

Reviewer #1: No

Reviewer #2: **Yes: **Ioana Bouros

Reviewer #3: No

Reviewer #4: No

Reviewer #5: No

Figure Files:

While revising your submission, please upload your figure files to the Preflight Analysis and Conversion Engine (PACE) digital diagnostic tool, https://pacev2.apexcovantage.com. PACE helps ensure that figures meet PLOS requir

---

## [Decision Letter · Decision Letter 1]

1 Jul 2024

Dear Ms Liu,

Thank you very much for submitting your manuscript "RtEstim: Time-varying reproduction number estimation with trend filtering" for consideration at PLOS Computational Biology. As with all papers reviewed by the journal, your manuscript was reviewed by members of the editorial board and by several independent reviewers. The reviewers appreciated the attention to an important topic. Based on the reviews, we are likely to accept this manuscript for publication, providing that you modify the manuscript according to the review recommendations.

See comment by reviewer 2 regarding the package name. Once this has been addressed, we should be able to accept the manuscript without further review.

Sincerely,

Claudio José Struchiner, M.D., Sc.D.

Academic Editor

PLOS Computational Biology

Virginia Pitzer

Section Editor

PLOS Computational Biology

See comment by reviewer 2 regarding the package name.

Reviewer's Responses to Questions

**Comments to the Authors:**

Reviewer #1: All of my comments have been satisfactorily addressed. I particularly appreciate the thoughtful evaluation and discussion of confidence intervals for Rt.

Reviewer #2: In this manuscript the authors have developed a frequentist approach for estimating the instantaneous reproduction numbers by transforming the process in to an optimisation problem with an added L1 penalty term to ensure smoothing of estimates, i.e. R(t-1) = R(t). Using cross-validation, the method allows for automatic fine-tuning of the hyper parameter \\lambda that scales the penalty term. The method also allows for fine-tuning the \\lambda and k (smoothing of results using kth splines), by minimising the KL divergence. The newly-developed approach is then compared to other top-ranking methods for fast inference of Rt trajectories for a set of both synthetic and real data for two different diseases in a clear and rigorous manner.

This new version of the paper has sufficiently tackled all the concerns raised by the previous reviewers. One small suggestion is that there should perhaps be a consistency of naming of the package: it is interchangeably referred as both RtEstim and rtestim.

Reviewer #3: The authors have addressed my concerns thoroughly.

Reviewer #5: The updated manuscript has address the questions I raised.

**Have the authors made all data and (if applicable) computational code underlying the findings in their manuscript fully available?**

Reviewer #1: Yes

Reviewer #2: Yes

Reviewer #3: Yes

Reviewer #5: None

PLOS authors have the option to publish the peer review history of their article (what does this mean?). If published, this will include your full peer review and any attached files.

Reviewer #1: No

Reviewer #2: No

Reviewer #3: **Yes: **Laura F White

Reviewer #5: No

Figure Files:

Data Requirements:

Reproducibility:

References:

---

## [Editor Report · Decision Letter 2]

15 Jul 2024

Dear Ms Liu,

We are pleased to inform you that your manuscript 'rtestim: Time-varying reproduction number estimation with trend filtering' has been provisionally accepted for publication in PLOS Computational Biology.

Best regards,

Claudio José Struchiner, M.D., Sc.D.

Academic Editor

PLOS Computational Biology

Virginia Pitzer

Section Editor

PLOS Computational Biology

---

## [Editor Report · Acceptance letter]

1 Aug 2024

PCOMPBIOL-D-23-02061R2 

rtestim: Time-varying reproduction number estimation with trend filtering

Dear Dr Liu,

I am pleased to inform you that your manuscript has been formally accepted for publication in PLOS Computational Biology. Your manuscript is now with our production department and you will be notified of the publication date in due course.

With kind regards,

Anita Estes
